# Looking for Visitor’s Effect in Sanctuaries: Implications of Guided Visitor Groups on the Behavior of the Chimpanzees at Fundació Mona

**DOI:** 10.3390/ani9060347

**Published:** 2019-06-13

**Authors:** Jana López-Álvarez, Yaiza Sanjorge, Sara Soloaga, Dietmar Crailsheim, Miquel Llorente

**Affiliations:** 1Innovació i Formació, Fundació Universitat de Girona, 17003 Girona, Spain; janalpzalv@gmail.com (J.L.-A.); yaizasc95@gmail.com (Y.S.); sarasoloaga@gmail.com (S.S.); 2Unitat de Recerca i Etologia, Fundació Mona, 17457 Girona, Spain; mllorente@fundacionmona.org; 3Facultat d’Educació i Psicologia, Universitat de Girona, 17004 Girona, Spain; 4Institut de Recerca i Estudis en Primatologia—IPRIM, 17005 Girona, Spain

**Keywords:** welfare, visitor effect, sanctuary, captivity, behavior, human interaction, chimpanzee

## Abstract

**Simple Summary:**

Organizations housing wildlife have a great potential to raise awareness and to contribute to conservation causes, along with a great responsibility towards the animals in their care. Displayed animals, taking on the role of ambassadors, are often exposed to the influence and actions of a great number of non-familiar humans. Studies trying to quantify the impact of visitors on captive housed animals have been very contradicting. In this study we report a neutral visitor impact on the behavior of chimpanzees, housed at a sanctuary with strict visitor protocols and supervision as well as animal management strategies, allowing animals a certain amount of choice and control over their visibility. By contrasting the mild visitor impact observed at the sanctuary to a great many studies, conducted at zoos with unsupervised free roaming visitors, often reporting undesirable effects, we wish to emphasise the importance to carefully managed visitor activities. We suggest that it is possible to mitigate potentially harmful visitor effects by restricting and supervising the visitor’s freedom of actions as well as providing animals on display with the means to evade or at least cope with the presence of visitors.

**Abstract:**

The question of ‘if and how captive primates are affected by visitors’ has gained increasing attention over the last decades. Although the majority reported undesirable effects on behavior and wellbeing, many studies reported contradicting results. Most of these studies were conducted at zoos, typically with little or no control over visitors’ actions. Yet little is known about the impact under very controlled visitor conditions. In order to fill this gap, we conducted this study at a primate sanctuary which allows public access only via a guided visit under strict supervision. We observed 14 chimpanzees, recording their behavior during, after and in the absence of guided visits over a 10-month period. Furthermore, we categorized the visitors regarding group size and composition to see if certain group types would produce a stronger impact on the chimpanzees’ behavior. As expected, we found visitors at the sanctuary to produce only a neutral impact on the chimpanzees’ behavior, detecting a slight increase of locomotion and decrease of inactivity during visitor activities with chimpanzees demonstrating more interest towards larger sized groups. We argue that the impact has been greatly mitigated by the strict visitor restrictions and care strategies allowing chimpanzees a certain control regarding their visibility.

## 1. Introduction

The fascination towards wildlife as well as its exhibition can be traced throughout history right up to the present day [1]. Animals, especially exotic species, were and are a matter of great interest. Humans want to be able to see as well as learn about animals [2] with records of the existence of zoological collections dating back to the fifteenth century [3]. Primates, particularly great apes, have been one of the most exhibited and popular animals in zoos [4]. Several survey studies demonstrated that the attraction towards monkeys and great apes often surpasses that of other species [5,6,7]. Over the last few decades, an increasing number of zoos changed from being purely entertainment-orientated to a more modern approach with education, conservation and research as primary goals [8]. These new responsibilities might be achieved via educational activities, captive breeding programs and management of wildlife within the zoo installations, [9] or in some cases in-situ conservation efforts consisting of reintroduction programs or financial support of field conservation projects [10,11].

Zoos have the capacity to reach an enormous number of people and influence their perception of animals as well as inform about their needs and the dangers they face in their natural habitat, mostly caused by human activities. Regardless of the mission statement or general objective of an animal housing organization, their success depends greatly on the wellbeing and health state of their animal collection. Thus, while educating the public about the influence of human activities on wild populations, it seems logical to also have in mind the possible effects humans have on captive populations.

The conservation status of nonhuman primates, especially great apes, is critical. Almost 60% of primate species are classified as endangered, with all great apes species being listed as either vulnerable, endangered or critically endangered [12]. The latest reports from the Convention on International Trade in Endangered Species (CITES) state a global primate trade of 450,000 live animals during 2005–2014 [12]. Some reports estimate that 22,000 wild great apes were lost between 2005 and 2011 [13]. Furthermore, the number of primates confiscated from the pet and entertainment industry, laboratories and illegal trafficking continues to grow [14]. Due to health conditions, financial limitations, habitat destruction, land-cover changes and industry-driven deforestation, many of these confiscated animals are being transferred to zoological parks or animal rescue centers as their only option [15]. Others, often even legally obtained and housed, end up at zoos or sanctuaries after becoming a ‘surplus’, no longer useful for their original commercial purpose [16].

European and North American primate sanctuaries mostly take in primates used formerly in the biomedical research, entertainment and pet industries. One currently accepted definition of an animal sanctuary in the United States comes from the Captive Wildlife Safety Act (CWSA) of 2007 (‘Captive Wildlife’), which is specific to big cats. According to this document, a sanctuary is an accredited, non-profit institution that does not propagate, commercially trade, allow direct contact or breed with the animals in their care [17]. These requirements also form a part of the North American Primate Sanctuary Alliance (NAPSA) definition of a true sanctuary. NAPSA goes even further and adds: “animals are not removed from the sanctuary for exhibition, education, research, or commercial purposes; public visitation is limited; animals are not trained to perform; the organization is fiscally responsible with a goal of providing lifetime care for sanctuary residents; and the sanctuary advocates for the species they care for” [18]. The European Alliance of Rescue Centres and Sanctuaries (EARS) uses a similar definition, uniting non-governmental & non-profit organization (NGOs) dedicated to the rescue and rehabilitation of a variety of species, not only limited to primates.

Nowadays, wildlife sanctuaries not only house and care for rescued animals, but also strive to advocate for improved captive animal welfare conditions, law enforcement, raising awareness and in-situ conservation [19]. Some sanctuaries, such as the Fundació Mona, additionally conduct non-invasive research focused on animal welfare [20] and education programs in the form of guided visits and academic programs [21]. These efforts to share information with the public and professionals are orientated towards reducing harm and threats to captive and wild primate populations. Yet some entities argue against displaying rescued animals to the public, due to a concern about the potential negative impact on the welfare of these animals. These concerns are addressed in the more common sanctuary definitions mentioned above, with remarks such as “not to use animals for commercial purposes” or “only limited public visitation”.

Behavioral studies related to human–animal interactions have become more popular since the 1970s, typically as attempts to quantify the welfare impacts of humans on zoo animals [22,23]. Many such studies focused on primates housed in a zoo setting, and reported contradicting results [24]. Whereas some studies concluded that visitor presence increases the animals’ stress levels [25] and leads to behavioral changes such as increased agonistic or stereotypical behaviors and decreased intra-group affiliations and exploration [26,27,28,29], others suggest human–animal interactions to have a positive effect functioning as environmental enrichment [30]. Some of these contradictions might be explained by studies suggesting that the response to human presence may vary among individuals [31,32] and can be mediated by personality [33] and thus can be difficult to detect.

Considering the high amount of attention this topic received in zoo settings, the lack of information on sanctuary housed animals is surprising and regretful. Primate sanctuaries aim to rehabilitate, and socially integrate rescued primates, by offering an environment suitable to express species-typical behaviors, allowing for a slow recovery and introduction in an adequate social network. As such, it seems extremely important to assess whether non-familiar human–animal interactions can delay, disrupt or distort subject’s rehabilitation and their state of wellbeing. In zoos as well as some primate rescue centers, animals are confronted with both familiar (caregivers, volunteers and researchers) and unfamiliar humans (visitors).

However, differences in the enclosure designs and visitor access strategies are expected to influence the impact of visitors on animals greatly. Enclosures can be designed to maximize the possibility of spotting animals to increase the visitor’s satisfaction or to allow animals to easily retreat to off-display areas avoiding a forced on human presence. Visitors might be allowed to roam freely without any supervision or can be restricted to a guided visit under strict supervision [34,35]. Visitors might be encouraged to feed or interact with animals while other organizations might follow a strict no human–animal interaction protocol [36]. We expect that such factors influence the effects non-familiar human presence might have on the animals and believe this could be a possible explanation for the controversial results from previous studies.

With much information already available from zoos, with little to unrestricted visitor access towards the animals, the aim of this study is to gain insights on the human impact in a sanctuary setting, being typically much more concerned about human interferences.

At Fundació Mona, visitors have a very restricted access to the animal, favoring the animal’s privacy over the visitor’s freedom of movement and action. Animals are exposed for only a limited number of hours a day with all visitor groups being guided and following strict behavioral protocols to minimize non-familiar human influence.

The general objective of this study is to test if and how the visitor presence has an impact on the chimpanzees housed at Fundació Mona, considering its very controlled visitor activity. For that purpose, we split our observation data depending on the visitor presence as “absence of visits”, “during a visit” and “after a visit”, expecting to find only mild alterations in their behaviors. Furthermore, we want to assess if certain aspects of visitor groups increase a possible impact on the chimpanzees behavior, by taking group size and composition into account.

## 2. Materials and Methods

### 2.1. Study Site and Animals

This study has been conducted at Fundació Mona (www.fundaciomona.org) a primate rescue center located in Riudellots de la Selva (Girona, Catalonia, Spain). Fundació Mona is member of the European Alliance Rescue Centres [19] dedicated to the rescue and rehabilitation of chimpanzees (*Pan troglodytes*) and Barbary macaques (*Macaca sylvanus*) since 2001. At the time of this study, a total of 14 chimpanzees were housed in two social groups at the center. Each group had a habitat consisting of an outdoor and two to three indoor areas. At night time, chimpanzees were confined to the indoor areas while during day time had either access to all areas or were confined to the outdoor areas, depending on the weather conditions and maintenance activities.

The chimpanzee outdoor area consists of two separate but adjunct enclosures, measuring 2420 m^2^ and 3220 m^2^ respectively, with a total perimeter of 191 m. The enclosures are surrounded by a steel fence and electrified wires for security reasons. Both enclosures were equipped with climbing structures, such as wooden platforms, towers and other structures as well as climbing ropes, enrichment devices, hammocks and ad libitum water dispensers. The outdoor area was a naturalistic environment maintaining the original ground substrate and Mediterranean vegetation. In order to provide certain privacy for the chimpanzees, a thick wall of vegetation mostly consisting of bamboo (*Phyllosta chysaurea*) has been planted around the enclosures limiting the visitor visibility to two open space and three “hiden” viewpoints (Figure 1). All visitor areas are separated by at least a two meter distance from the animal fencing, making physical interactions impossible.

All 14 chimpanzees (*Pan troglodytes*) housed at the sanctuary, split in two mixed-sex groups (Bilinga group 4M3F, Mutamba group 5M2F), have been observed for this study (Table 1). Most of the primates housed at the center were previously held as pets or used in the entertainment industry before being confiscated and handed over to Fundació Mona. New rescues are being housed separately from conspecifics for several weeks before starting their integration into social groups. The time needed for the adaptation to their new home, the physical rehabilitation and social integration differs for each individual. Initial treatment and care protocols are specifically developed for each individual. A strict hands-off policy is being maintained at the center, with physical contact only allowed for a few qualified staff members for rehabilitation and veterinary purposes.

The chimpanzees were fed at least four times per day and water was provided ad libitum in the outdoor and indoor enclosures. The diet consisted mainly of seasonal vegetables and fruits, boiled rice, a variety of dried fruits, seeds and some protein-rich food items. A big portion of their diet was scattered and hidden in the outdoor areas in order to encourage foraging behaviors. Animals were routinely environmentally stimulated through feeding [37], social [38], sensorial [39] and cognitive [40] enrichment. Interactions with familiar humans (staff members) were maintained at a minimum, with care givers approaching the animals only during feeding times or care management activities in order to not interrupt but encourage intra-group interactions.

### 2.2. Visitor Strategies Applied at the Study Site

At the sanctuary only guided visits were conducted to give the public the possibility to see the chimpanzees while keeping possible disturbance for the primates to a minimum. Free roaming or unsupervised visitors were strictly forbidden. During the duration of this study organized scholar groups would visit the center only on week days while family and adults could participate in joined guided visits throughout the weekend. All visitor activities were limited to the morning hours and finished before 14:30. Although a visit could last between two to four hours, visitors would only spend about 1–1.5 h in areas close to the animals, potentially influencing the chimpanzees. All other parts of the visit, such as presentations or workshops were conducted either indoors or far off the animal areas. Trained visitor guides would inform participants before entering areas close to the animals about behavioral rules, including (1) restriction to cross any barrier in order to approach the animal installations; (2) not to try to interact, call the attention, talk to or disturb the animals in any way; (3) restriction of not eating, drinking or smoking in sight of the animals; and (4) trying to feed any animals during the tour. The main tasks of a visitor guide were to educate visitors about the needs, threads and dangers wild and captive primates are currently facing as well as to assure that visitors would abide to the visitor protocol.

### 2.3. Data Collection

Data on their behavior were collected only in the outdoor areas once access was granted to the chimpanzees, between 10:00 AM and 15:00 PM (slight variation due to weather conditions). We excluded observations between 15:00 PM and 19:00 PM as more than 90% of this data were labelled as “in absence of visit” and chimpanzees might act differently in the afternoon compared to the morning hours independently from visitor activities. Data were recorded from observation towers located between the outdoor enclosures and the visitor walk-around. Chimpanzees were already habituated to the presence of observers due to ongoing monitoring projects. Data on the chimpanzees’ behavior and visitor presence were recorded from March 2018 until January 2019, resulting in 38,700 recorded behaviors, excluding when animals were not visible or the behavior was obscured (average of 2766 ± 710 behaviors per individual). Observations were equally distributed throughout the week and across the morning hours. Only data from trained observers were used for this study after successfully passing the inter-observer reliability test (agreement ≥ 85%) with the head of research at the center (M. Llorente).

Data were coded using the instantaneous scan sampling method [41] every two minutes on all chimpanzees present of one group for 20 min. Observers used tablets with the ZooMonitor data scoring software [42] programmed with the sanctuaries monitoring ethogram, consisting of three visitor-related (Table 2) and 18 behavior-related (Table 3) categories.

Based on the behavioral records we calculated two welfare indices previously used by Llorente et al. [20]. We calculated the Behavioral Competence Index (BCI), which contrasts individual desirable behaviors (positive) against individual not desirable behaviors (negative in excess), using the following formula:$$\mathrm{BCI}\;=\;\frac{\mathrm{Positive}\;\mathrm{individual}\;\mathrm{behaviors}-\mathrm{Negative}\;\mathrm{individual}\;\mathrm{behaviors}}{\mathrm{Positive}\;\mathrm{individual}\;\mathrm{behaviors}+\;\mathrm{Negative}\;\mathrm{individual}\;\mathrm{behaviors}}$$

We considered feeding, locomotion, manipulation and individual play as positive individual behaviors; and abnormal, inactivity and self-directed as negative individual behaviors. Although inactivity and self-directed behaviors belong to the normal chimpanzee ethogram, we considered a high frequency of these behaviors as not desirable.

We calculated the Social Preference Index (SPI) which contrasts the sum of all individual behaviors against all social behaviors, using the following formula: $$\mathrm{SPI}=\frac{\mathrm{Social}\;\mathrm{behaviors}-\mathrm{Individual}\;\mathrm{behaviors}}{\mathrm{Social}\;\mathrm{behaviors}+\mathrm{Individual}\;\mathrm{behaviors}}$$
values of both indices range between −1 and +1. Similar indices have proved to be useful in other behavioral studies [43].

The definition for the behavior “human interaction”, having a somehow misleading name, does not refer to humans interacting with the chimpanzees, but rather the chimpanzees exhibiting one of the following reactions: (1) agonistic display directed towards location of visitor group; (2) following the visitors and their approximate trajectory (while out of sight); (3) approaching the fence closest to the visitors directing attention towards location of visitor group; or (4) calling visitors attention.

### 2.4. Statistical Analysis

We converted the absolute frequencies of behaviors into ratios, whilst excluding the “not visible” entries from the total frequency of observed behaviors. We than calculated the data points for the statistical analysis based on the observed behaviors for each chimpanzee per contrasted visitor parameter. For the first set of analysis we obtained three values for each behavior per chimpanzee and visitor condition (none, visit, after). For the second set of analysis we further included the visitor group aspects (size, type). However, in order to maintain a high quality, we excluded all data points from this study that were calculated with less than 40 recognizable behaviors.

To demonstrate the general impact visitors have on the animals we used Statistical Package for the Social Sciences (SPSS) to compare chimpanzees behaviors (inactivity, locomotion, intra-group affiliative and intra-group agonistic interactions, abnormal & self-directed behaviors) and welfare indices (SPI and BCI) during, after and in the absence of visitors. Friedman ANOVA and Wilcoxon signed rank test have been applied with an alpha level of <0.05 to evaluate these effects.

Furthermore, we ran generalized linear mixed models (GLMMs) using the “lme4” package [44] in R, using only records of the chimpanzees behavior during and after visitor presence in order to gain a better understanding on which specific aspects of the visitor groups cause alteration of the chimpanzee’s behavior. We ran normal GLMMs with an identity link function after visually inspecting the residuals normal distribution in the QQ plots (Appendix A). GLMMs were created with the same model composition (random and fixed factors), only changing the depended variable for each model. We used the individuals “ID” and “Group” as random factors (Table 1) and “Visit condition”, “Visit group type” and “Visit group size” as fixed factors (Table 2). We included “Group” as a random factor, suspecting that the location and/or design of the chimpanzee group’s respective installations might influence the visitor impact. Model 3 with human interaction as a dependent variable was the only exception, which excluded “Visitor condition” as a human interaction towards visitors could not occur “after a visit” and would result in a false significance. 

We chose to use SPI, BCI, human interaction, intra-group affiliative and agonistic interactions, locomotion, inactivity and abnormal & self-directed behaviors as dependent variables. In all models we tested that the full models, containing all fixed factors, were significant improvements over the null models, without fixed factors, by applying the likelihood ratio test. When full models differed significantly from the null models, we applied likelihood tests on the full model to obtain a p-value for each fixed factor by using the R function ANOVA (Satterthwaite’s method) [45]. For the fixed factor “Visit group size” with three categories (small, medium, big) we applied a multiple comparison of means post hoc test with Tukey Contrast and Holm–Bonferroni p-value adjustment to verify the significance between the categories.

We tested for multicollinearity between all fixed factors by calculating the variance inflation factor (VIF) using the “car” package in R [46]. All VIFs calculated for our three fixed factors were below 1.1, indicating that our fixed factors were not correlated.

### 2.5. Ethical Note

This research was conducted in accordance with all national and intuitional guidelines for the care and management of primates established by Fundació Mona, Association for the Study of Animal Behavior Society and the Spanish Government (RD 53/2013).

## 3. Results

As can be observed by looking at the differences between the behavioral budgets calculated, first globally then for each visitor condition (none, visit, after), no drastic alterations were detected. We included *“*not visible*”* (described as either *“*behavior obscured*”* or *“*individual not visible*”*) in the graphic representation (Figure 2) in order to demonstrate the capacity of the chimpanzees to choose not to be seen by humans (familiar and non-familiar alike). However, we did not include *“*not visible*”* in any of the statistical analysis, as we were often unable to discern if chimpanzees chose to spend time in areas not visible to visitors alone or visitors and observers at the same time (due to differences in standpoint from which observers and visitors watched the chimpanzees).

### 3.1. Impact of Visitor Presence on Chimpanzee Behavior

For this analysis, we compared the dependent variables between the three possible conditions of visitor presence ((Table 4) none, visit, after). We obtained significant differences in inactivity (*χ*^2^ = 11.286, *p* < 0.01), locomotion (*χ*^2^ = 9.571, *p* < 0.01) and agonistic intra-group interactions (*χ*^2^ = 6.167, *p* < 0.05). Chimpanzees were significantly less active after a visit, compared to during a visit and spend more time on locomotion during a visit than in the absence of visitor groups. Although agonistic intra-group interaction scored a significant result, we could not detect any clear differences between the three categories, which might be due to the relatively infrequent occurrence and high individual variations between individuals (Figure 3).

The BCI and abnormal and self-directed behaviors did not seem to be affected by the visitor condition, while the SPI and affiliative intra-group interactions, although not producing any significant results, exhibited certain tendencies of decreased social activities during visits compared to after as well as during the absence of a visit.

### 3.2. Influence of Visitor Group Aspects on the Chimpanzees Behavior

Here we intended to demonstrate that specific aspects like the size and the type of visitor groups cccmight alter the magnitude of the impact during and after visits. We decided to use GLMMs in order to control for repeated observations of the same individual under different visitor conditions and aspects, as well as the possibilities that the groups could receive more or less visitor attention depending on the enclosure locations. We created a total of eight models (Table 5), but only two models (3 and 5) proved to be significantly better fits in order to help explain the possible alterations of the depend variables.

After discarding all models that failed to show a significant improvement to explain de dependent variables (Table 5), we continued further statistics only with models 3 and 5. We found significant differences in locomotion for size and type (Table 6) with chimpanzees engaging significantly more in locomotion with smaller than bigger visits (Table 7) as well as increased locomotion with family compared to scholar visits (Figure 4).

## 4. Discussion

Humans in the past and today have a great desire to see animals up close. Zoos and sanctuaries facilitate a quick and easy way to do so, without the need for lengthy and costly travels in order to see them in their natural habitat. This attraction might originate from personal curiosity, the simple wish to be entertained or the desire to learn about animals and possibly support causes to help specific individuals or species [47,48]. Regardless of the reason for visiting, modern zoos and sanctuaries have the potential and responsibility to educate visitors and actively raise awareness about the needs and threats many species currently face, converting visitors into part of the solution rather than being part of the problem. For the great majority of endangered species, like chimpanzees, this includes explaining that human actions pose the biggest threat [49]. However, allowing public access to animal collections and converting these animals into ambassadors for animal welfare and conservation causes, does not mean that their own wellbeing can be ignored. On the contrary, it should be compensated for by an additional effort to provide the best care possible and shield them from potentially negative stimulations.

Many aspects of a captive environment need to be controlled, designed and well organized to provide a high level of welfare [50,51]. The more actively used or present an element or aspect in the chimpanzee’s day to day life, the more important it seems to monitor and, if necessary, control its effects on the animals. Thus, we argue that the effects of non-familiar human presence (visitors) should be monitored and regulated.

If we summarize the results of this study in simple terms of “how visitors affected the chimpanzees housed at the sanctuary”, we would report the impact as neutral or ambivalent, with no clear indications of any positive or negative orientated alterations in their behavior budget and in extension wellbeing. This study has been conducted on purpose at a sanctuary with very strict care and welfare strategies as well as a very controlled and restricted visitor access, favoring the animal’s privacy and mediation of visitor effects. Thus, we expected to find little to no significant alterations in their behavior, which has been confirmed by our data.

Although the visitor activity at our study has to be considered a very controlled condition, we wish to emphasize the possibility of individual differences regarding the animals’ perceptions and in continuation reaction to stimulations such as visitor presence. These differences, detectable as variations in intensity of interest in visitors and stronger/weaker changes to their behavior budget might be explained by the individuals’ personality [33,52] or past traumatic experiences [53,54,55]. While in this study we did not wish to investigate individual differences, we were aware of this possibility and thus controlled these through the use of random factors (ID, group) in our GLMMs, to assure that our results would not be affected. This seemed especially important due to our relatively small sample size of only 14 chimpanzees in two groups.

In our first set of analysis, comparing the chimpanzee’s behavior budget between the absence of visitors, while being exposed to visitors and shortly after visits, only three behaviors were affected. Locomotion increased during visits compared to absence of visitors and inactivity decreased during visits compared to after a visit. Activity levels are frequently reported to change in visitor impact studies, especially in primates [56]. Hosey reported locomotion and inactivity to increase drastically in several studies of different primate species [57]. Others argue that an increase in locomotion should be taken seriously as it could be a first step towards developing stereotypical behaviors such as pacing [58]. However, we found only an average increase of 1.8% in locomotion and average decrease of 4.8% in inactivity. Still, even a small increase in locomotion could be argued to be a sign of agitation. Nevertheless, as locomotion in our ethogram only represents the change of location, not including pacing (which was recorded as abnormal behavior in this study) or human interaction (animals moving or positioning themselves close to or following the trajectory of visitor groups), we would not identify this as an agitation-triggered response. We argue that neither of those behavioral alterations would be considered an extreme reaction or would suggest a clear negative or positive impact on the animal’s wellbeing. Intra-group agonistic interaction was the third behavior initially detected as significantly affected. However, the following Wilcoxon signed rank test failed to confirm any significant difference between during, after or in absence of visits. This false positive was most likely caused by the extremely low occurrence of agonistic behaviors with only 0.3% ± 0.5% of their behavior budget and the fact that several animals were never observed to engage in agonistic behaviors during and after visits.

Regarding the analysis of visit specific factors “visit group size” and “visit group type” during and after visits, only two out of eight models were significant improvements over the respective null models. This implies that none of our recorded visitor parameters were suited to explain any changes in the SPI and BCI, inactivity, affiliative and agonistic behaviours or abnormal and self-directed behaviors. Hence, we have to assume that the effects of the visitor group attributes were either too mild to be detected in our sample or that other variables, not taken into account in this study, might have stronger influences. Only the full models of locomotion and human interaction were significantly better fits than their corresponding null models. Locomotion increased significantly during family visits compared to scholar visits and decreased significantly while being exposed to big sized visits (more than 30 people) compared to small sized visits (less than 15 people). As mentioned before, the increase of locomotion to the small extent found in our sample does not indicate any welfare related problems. However, we were surprised to find that this increase occurred due to small sized visitors groups, considering how most studies typically reported bigger crowd sizes provoked increased wounding [59], elevated stress levels [60] or undesirable alterations of behaviors [61].

The human interaction model, only containing “visitor group size” and “type” as fixed effects, showed a significant increase during big visitor groups. This indicates a stronger attraction or interest in humans when group size exceeded a total of 30 people. This could be considered an interruption of their normal routine. However, considering that human interaction behaviors only made up 3.5% of the chimpanzees’ activity budget during visits, this should be regarded a minor alteration of their activity budget. Furthermore, out of all observed interactions from chimpanzees towards visitors, only 8.1% were categorized as negative, mostly consisting of agonistic displays directed at visitors, while all others were labelled as neutral/positive, consisting of approaching and following the visitors or closely observing visitor groups. Nevertheless, the fact that human interaction behaviors augmented with increasing group size coincides with observations on gorillas [62] and orangutans (*Pongo borneo*) [63] who exhibited stronger reactions towards bigger compared to smaller crowds at zoos.

We believe to have evidenced that a sanctuary can be open to the public, with rescued chimpanzees as ambassadors to promote animal welfare and conservation without necessarily affecting the chimpanzee’s wellbeing if certain conditions, such as adequately restricted visitor strategies carefully designed enclosures and appropriate care management strategies, are met.

## 5. Conclusions

Davey pointed out in a review about visitor effects how contradictory the results of past studies were. Although the majority of studies conclude visitor effects to be either stressful or at least ambivalent, it seems clear that more aspects have to be taken into account [24]. While it seems obvious to expect differences between species [64] or even individuals, we want to emphasise the importance of parameters that are being controlled by the organizations. This includes restricting visitor’s freedom of movement and actions, and providing the animals with means to avoid visitor presence. We believe that, in many cases, contradicting study results could be explained by looking at how the visitor activities were organized and which strategies were used to buffer and reduce potential visitor effects. Thus, we deduce that the visitor impact turned out to be very mild at Fundació Mona due to the following strategies applied by the sanctuary:
(1)Allowing the animals a certain amount of choice towards de-visibility, by creating off-display areas and maintaining an adequate buffer distance between enclosures and visitor areas. Blaney and Wells described a reduction of aggressive and abnormal behaviors after installing a camouflaged netting around a gorilla (*Gorilla gorilla*) enclosure [65]. Several studies reported favourable improvements after remodelling enclosures with the goal to provide animals with retreat possibilities [66]. A study on cotton-top tamarins (*Saguinus oedipus*) reported significant improvements in social behaviors for groups being housed off-display compared to groups being visible to zoo visitors [67]. At our study site, a dense wall of vegetation was installed on long stretches around the enclosure to reduce visibility (both ways). Furthermore, enclosures were designed in a way to allow chimpanzees to retreat without being seen and to keep their distance from visitors if desired.(2)Restricting and supervising visitor’s movements and actions, making active intents to interact or react to animals impossible. To our knowledge most studies have been conducted in zoos with visitors roaming freely and unsupervised. To us, it seems that this might be the biggest difference between our study site compared to most studies that have reported negative reactions. This would suggest that the quality might be more important than the quantity [61]. With quality we refer to the visitor’s attitude towards the animals, in terms of acting respectful, abiding visitor regulations, attempting not to disturb or startle animals. In addition to the strict visitor protocol put in place at Fundació Mona, the visitor guides have the important task of passively and actively preventing visitors from disturbing or attempting to interact with the chimpanzees. Parker et al. tested [68] the efficiency of “do not feed” signs at zoos and reported that while animal feeding decreased, other behaviors such as attempts to touch the animals increased at the same time. This study showed that visitors, although possibly being informed not to disturb or interact with the animals, when allowed to move freely and unsupervised close to the habitats will most likely cause disturbances for the animals. That being said, it might have been interesting to also collect information on the age of visitors, the ratio between adults and children in each group and especially taking the presence of toddlers and infants into account. While the majority of children in scholar groups ranged between the age of four and 12, with groups consisting nearly entirely of children, the adult to children ratio in family groups could vary greatly and children of all ages could be present (including <4 years of age). Assuming that younger visitors and toddlers are less likely to understand and follow strict behavioral rules and are more likely to produce sporadic loud noises or unpredictable movements which could potentially startle animals, they might affect animals more strongly. As such information was not recorded in our observation phase, we have no clear results confirming this suspicion, but wish to state this as one possible explanation to why locomotion increased more strongly during family groups than scholar groups. Accordingly, we suggest controlling for visitor age and likelihood to comprehend and follow visitor guidelines in future study designs.


We argue that zoos and sanctuaries have a great potential to influence people, raise awareness and promote conservation programs, but strongly advise strict monitoring and regulation of visitor activities to ensure the wellbeing of the displayed animals.

The visitor impact on exhibited animals depends greatly on both the actions of the visitors as well as the capacity and possibilities of the animals to control their environment and cope with the situations. That being said, it is the sheltering organization that has the power to restrict and guide visitors’ actions and it has to offer the animals a certain amount of choice and control.

Although this study was conducted only on chimpanzees, we are convinced that our conclusions would be valid for a variety of other species as well. Thus, we hope and strongly suggest a reproduction of this study with other species in the future.

## Figures and Tables

**Figure 1 animals-09-00347-f001:**
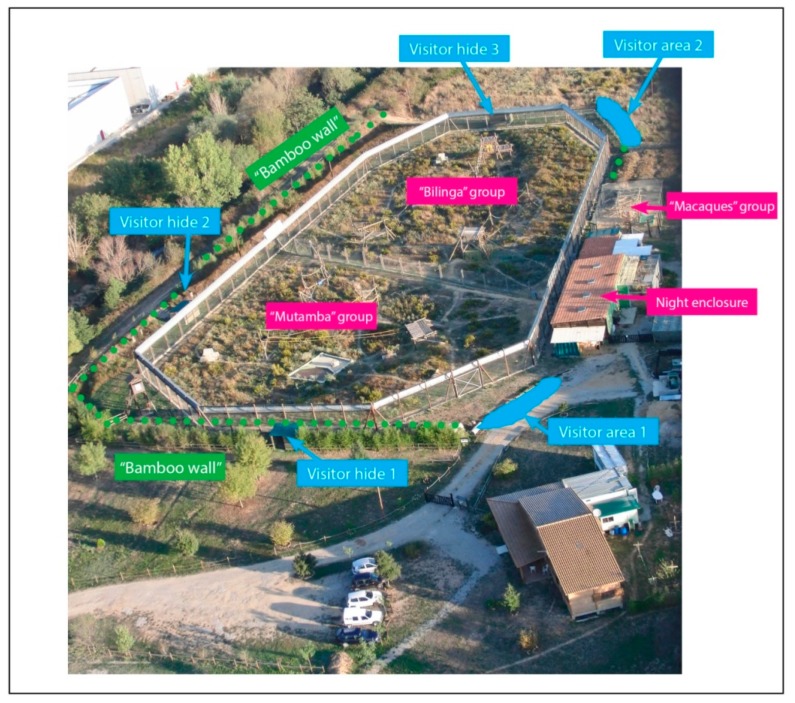
View of the animal facilities and visitor areas at Fundació Mona: Animal habitats labelled in pink; Visitor viewpoints labelled in blue; Bamboo visibility block labelled in green.

**Figure 2 animals-09-00347-f002:**
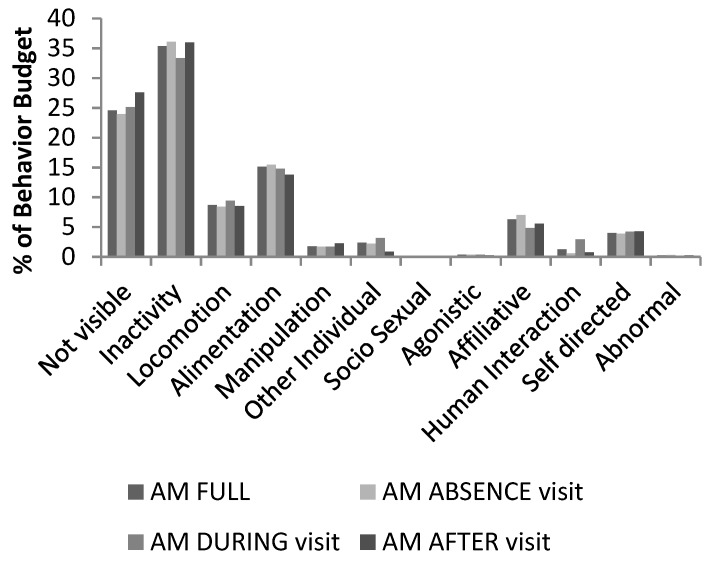
Chimpanzee behavior budget, based on the observation records (AM: from 10:00–15:00) used in this study. Behavior budget were calculated independently first using all available data (AM FULL), but also calculated for each Visitor condition (none, visit, after), to allow a visual quick overview.

**Figure 3 animals-09-00347-f003:**
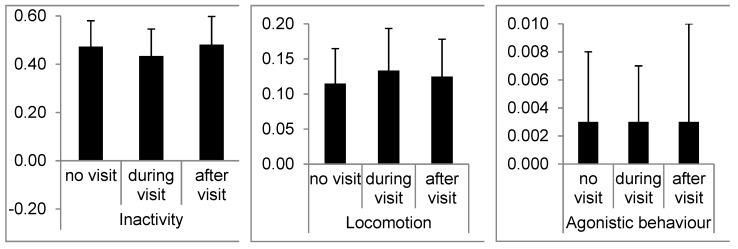
Averages of inactivity, locomotion and intra-group agonistic behavior according to visitor presence (none, visit, after).

**Figure 4 animals-09-00347-f004:**
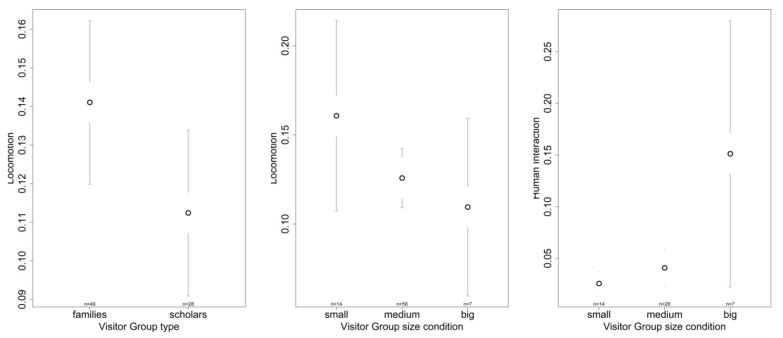
Behaviors (locomotion and human interaction) of the chimpanzees significantly affected by aspects of the visitor groups with a confidence interval of 95%.

**Table 1 animals-09-00347-t001:** List of biographical information on all chimpanzees housed at Fundació Mona.

Name	Gender	Birth Year	Arrival at Mona	Group
Bongo	Male	2000	July 2002	Mutamba
Waty	Female	1997	June 2002	Mutamba
Marco	Male	1984	March 2001	Mutamba
Charly	Male	1989	March 2001	Mutamba
Africa	Female	1999	May 2009	Mutamba
Toni	Male	1983	August 2001	Mutamba
Juanito	Male	2003	January 2005	Mutamba
Bea	Female	1985	May 2012	Bilinga
Coco	Female	1994	May 2012	Bilinga
Victor	Male	1982	May 2006	Bilinga
Nico	Male	2001	March 2004	Bilinga
Tico	Male	1987	July 2005	Bilinga
Cheeta	Female	1990	November 2015	Bilinga
Tom	Male	1985	June 2011	Bilinga

**Table 2 animals-09-00347-t002:** List of parameters recorded for visitor group categorization.

**Visitor Related Variables**	**Visit Group Type ^1^**	**Families, Scholars**
	Visit group size ^2^	Small (<15), Medium (15–30), Big (30–52)
	Visit condition ^3^	No Visit, During a visit, After a visit

^1^ Group type differentiates between organized “Scholars” groups consisting of pupils with a small number of adult patrons and “Families” consisting of Families and adults that are unfamiliar to each other. ^2^ Group size intervals are based on the tendency of the sanctuary of organizing visits, typically having one guide for up to 30 participants and trying to avoid visitor groups of more than 30 people. ^3^ Condition is based on the presence of visitors at the viewpoints located around the enclosures and “after a visit” was defined as the time period up to 20 min (full observation session) after the visit group left the observed groups enclosure surrounding.

**Table 3 animals-09-00347-t003:** List of behaviors recorded using two minutes instantaneous scan sampling.

Behavior Variables	Individual	Inactivity, Feeding, Locomotion, Manipulation, Self-Directed, Solitary Play, Abnormal, Human Interaction, Other Individual
	Social	Grooming, Social Play, Other Affiliative, Agonistic Dominance, Agonistic Submission, Other Agonistic, Socio Sexual
	Other	Not visible

**Table 4 animals-09-00347-t004:** Dependent variables affected by the visitor presence condition, comparing the three sub categories (none, visit, after).

			Wilcoxon Test					
	Friedman Test		None-Visit		Visit-After		None-After	
Dependent Variables	Chi-Square	*p*	z	*p*	z	*p*	z	*p*
BCI	2.714	0.257	−	−	−	−	−	−
SPI	5.571	0.062	2.794	0.005 **	1.287	0.198	2.291	0.022 *
Inactivity	11.286	0.004 **	1.915	*0.056*	2.731	0.006 **	0.910	0.363
Locomotion	9.571	0.008 **	2.919	0.004 **	1.726	*0.084*	1.224	0.221
Affiliative	5.538	0.063	2.830	0.005 **	1.572	0.116	1.992	0.046 *
Agonistic	6.167	0.046 *	0.245	0.807	0.652	0.515	1.503	0.133
Abnormal & Self-Directed	1.714	0.424	−	−	−	−	−	−

* *p* ≤ 0.05, ** *p* ≤ 0.01.

**Table 5 animals-09-00347-t005:** List of all generalized linear mixed models (GLMMs) and the results from the likelihood ratio test between the respective null and full models.

GLMM Model	Dependent Variable	Chi-Square	Df	*p*
**MODEL 1**	SPI	6.336	4	0.176
**MODEL 2**	BCI	5.447	4	0.244
**MODEL 3**	Human interaction	29.623	3	**0.000 *****
**MODEL 4**	Inactivity	6.684	4	0.154
**MODEL 5**	Locomotion	14.044	4	**0.007 ****
**MODEL 6**	Affiliative	7.654	4	0.105
**MODEL 7**	Agonistic	8.035	4	0.090
**MODEL 8**	Abnormal & Self-Directed	8.022	4	0.091

All models included the individuals ID and group as random factors and visitor condition (visit, after), visitor group size (small, medium, big) and visitor group type (families, scholars), with the exception of Model 3 (Human Interaction) which was excluding the fixed factor visitor condition. ** *p* ≤ 0.01, *** *p* ≤ 0.001.

**Table 6 animals-09-00347-t006:** Behavior of the chimpanzees significantly affected by aspects of the visitor groups.

				Fixed Factors (Visitor)		
GLMM	Random Factor	Dependent Variable		Condition	Size	Type
**MODEL 3**		**Human interaction**	F	−	15.101	2.658
	**ID, Group**		df	−	2	1
			*p*	−	**0.000 ****	0.108
**MODEL 5**		**Locomotion**	F	0.359	3.295	5.730
	**ID, Group**		df	1	2	1
			*p*	0.551	**0.044 ***	**0.020 ***

* *p* ≤ 0.05, ** *p* ≤ 0.001.

**Table 7 animals-09-00347-t007:** Post hoc test for comparison between sub categories of the visitor group size (small, medium, big).

			Predictors		
GLMM Model	Dependent Variable		Medium—Big	Small—Big	Small—Medium
**MODEL 3**	**Human Interaction**	z	−5.085	−5.175	−0.741
		*p*	**0.000 ****	**0.000 ****	0.459
**MODEL 5**	**Locomotion**	z	1.712	2.562	1.237
		*p*	0.174	**0.031 ***	0.216

* *p* ≤ 0.05, ** *p* ≤ 0.001.

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
