# Peer review of "Looking for Visitor’s Effect in Sanctuaries: Implications of Guided Visitor Groups on the Behavior of the Chimpanzees at Fundació Mona"

_animals, 2019, doi:10.3390/ani9060347_

Round 1

Reviewer 1 Report

Overall, this is a nice study that sets a precedent for work with animal sanctuaries. I can see something similar being done with other species (wolves, for one) in other parks. It is important that the welfare of non-human animals in our care to be considered. You address potential limitations nicely, as well. For example, noting the various ways increased locomotion during visitor presence could be construed as agitation, then addressing, with evidence, why you do not interpret the findings in such a way.

You may want to consider the possibility of this increased activity during family visits as a result of children present. Can you discuss the age range of the children/families? Young children, even if following the rules, make a different level of noise, move differently, and smell differently. These small variances could result in increased agitation or curiosity on the part of the primates being observed.

You should also note a limitation of the small sample and address any variations between the two groups observed (Mutamba and Bilinga). This could support personality differences, which you mentioned in the introduction.

Finally, a thorough copy edit for grammar, spelling, mechanics (multiple errors regarding capitalization), etc. It appears English is not the authors' first language, and for the most part, the translation is solid. There are places in which sentences could be revised for clarity and other grammar/spelling issues should be addressed.

Author Response

Dear Reviewer

thank you very much for taking the time and making the effort to help us refining this manuscript, hopefully becoming a valuable addition to already existing information regarding animal welfare and human influence on captive held animals. We are very pleased with the positive reception and thus highly motivated to keep working on this and similar topics in the future.

Reviewer 1 - Comments and Suggestions for Authors

Overall, this is a nice study that sets a precedent for work with animal sanctuaries. I can see something similar being done with other species (wolves, for one) in other parks. It is important that the welfare of non-human animals in our care to be considered. You address potential limitations nicely, as well. For example, noting the various ways increased locomotion during visitor presence could be construed as agitation, then addressing, with evidence, why you do not interpret the findings in such a way.

Answer: We are very pleased to hear that our view about the importance of this topic is being shared by Reviewer 1. We absolutely agree that more studies should be conducted including other species and other visitor settings to further highlight the importance of human visitor influence on captive held animals´ wellbeing.

Increased locomotion could be evaluated as something positive, if animals are generally observed to spend an undesired elevated amount of time inactive, or could be seen as a negative effect, arguing that it could indicate a certain agitation, produced by the presence of visitors. Depending on the habitat and study population we believe both arguments to be valid. Regarding our chimpanzees, we believe that the increase in locomotion seen during visits, particularly family groups, is not necessarily positive as the increase is not very high (average increase of 1.8%), assuming that the positive effect of increased locomotion would suggest an increase in activity. Neither would we interpret this result as negative, as locomotion does not include pacing (in our ethogram scored as abnormal behavior) or human interaction (animals moving or positioning themselves close to or following the trajectory of visitor groups), which could indicate an undesirable impact.

We now included such an explanation in the new manuscript and agree it makes a valuable addition to the discussion of our results.

You may want to consider the possibility of this increased activity during family visits as a result of children present. Can you discuss the age range of the children/families? Young children, even if following the rules, make a different level of noise, move differently, and smell differently. These small variances could result in increased agitation or curiosity on the part of the primates being observed.

Answer: We understand the base of this suggestion; however family groups would actually consist of fewer children (typically equal ratio or more adults than children) than scholar groups (all children except for a few teachers/monitors). However while the majority of children in scholar groups would range between the age of 4 to 12 years, family groups might include toddlers and children younger than 4 years of age as well.

We definitely agree that children are less inclined to strictly follow rules, or might not yet even understand them. Specifically the rule about avoiding sudden or loud noises and they tend to be less predictable in their actions and reactions. From ad libitum observations we would also suggest chimpanzees to show more curiosity towards children (especially younger ones of less than 4 years) but cannot provide any solid results confirming this observation with our data.

Regarding the smell, we agree that for many species the smell of visitors and particularly children can have a mayor effect on animals and should be taken into consideration. However chimpanzee's olfactory capacities are very similar to that of humans and it is quite unlikely that they would be able to perceive the visitors smell out in the open, even at the least distance allowed at the Mona site.

That being said we sadly did not record any information of the age or the proportion between adults and children in our observations. Nevertheless, as observations are ongoing, we will make sure to consider including records of such information for future publications.

We added a paragraph addressing this suggestion in the discussion/conclusion.

You should also note a limitation of the small sample and address any variations between the two groups observed (Mutamba and Bilinga). This could support personality differences, which you mentioned in the introduction.

Answer: We will add a paragraph in the discussion/conclusion about the small sample size and the fact that we observed two groups. By using Group and ID as random factors we were able to control these factors not to influence our global results, but we agree we should put more emphasis on the fact that we did this because we believe individual and group to be potentially affected differently. As reviewer 1 pointed out correctly, we mentioned it in the Introduction as well as the methodology but failed to address this in the discussion.

Finally, a thorough copy edit for grammar, spelling, mechanics (multiple errors regarding capitalization), etc. It appears English is not the authors' first language, and for the most part, the translation is solid. There are places in which sentences could be revised for clarity and other grammar/spelling issues should be addressed.

Answer: As the reviewers detected correctly, English is not our first language. We apologize for the grammar and language mistakes. We now had an English native speaker, colleague of ours, revise the manuscript and modified language concerns accordingly. That being said, if for some reason the English level would still not meet the journals standards, we would use the MDPI Author service.

Reviewer 2 Report

This is a well-executed study. Research design is appropriate, and data analysis is clearly presented, The conclusion is properly supported. Good writing. I found a few minor spelling errors. I hope you will go through the manuscript again and publish this work quickly.

Author Response

Dear Reviewer

thank you very much for taking the time and making the effort to help us refining this manuscript, hopefully becoming a valuable addition to already existing information regarding animal welfare and human influence on captive held animals. We are very pleased with the positive reception and thus highly motivated to keep working on this and similar topics in the future.

Reviewer 2 - Comments and Suggestions for Authors

This is a well-executed study. Research design is appropriate, and data analysis is clearly presented, The conclusion is properly supported. Good writing. I found a few minor spelling errors. I hope you will go through the manuscript again and publish this work quickly.

Answer: We are very pleased with the positive reception of the study topic as well as execution. As the reviewers detected correctly, English is not our first language. We apologize for the grammar and language mistakes. We now had an English native speaker, colleague of ours, revise the manuscript and modified language concerns accordingly. That being said, if for some reason the English level would still not meet the journals standards, we would use the MDPI Author service.
